# Data-Driven Position and Stiffness Control of Antagonistic Variable Stiffness Actuator Using Nonlinear Hammerstein Models

Ali Javadi [1], Hamed Haghighi [2], Khemwutta Pornpipatsakul [3] and Ronnapee Chaichaowarat [1,3,*]

1    International School of Engineering, Faculty of Engineering, Chulalongkorn University, 254 Phayathai Road, Pathumwan, Bangkok 10330, Thailand; alij63@gmail.com
2    West Azerbaijan Science and Technology Park, Urmia 333, Iran; hd.haghighi@gmail.com
3    Department of Mechanical Engineering, Faculty of Engineering, Chulalongkorn University, 254 Phayathai Road, Pathumwan, Bangkok 10330, Thailand; 6673002021@student.chula.ac.th
*    Correspondence: ronnapee.c@chula.ac.th

**Abstract:** In this paper, an optimal PID controller is introduced for an antagonistic variable stiffness actuator (AVSA) based on Hammerstein models. A set of Hammerstein models is developed for the AVSA using the voltage difference method. For each stiffness level, linear and nonlinear Hammerstein models are identified using the least squares method. Experimental results confirm that the outputs of the Hammerstein models fit the measured data better than linear models, as Hammerstein models can incorporate nonlinear effects such as friction. A genetic algorithm is utilized to find optimal PID gains for different stiffness levels and reference position amplitudes. The final gains are obtained by linearly interpolating the optimal gains obtained. To demonstrate the effectiveness of the proposed design, several scenarios with different reference positions and stiffness profiles are provided. Specifically, square, sinusoidal, and sawtooth waves are used for reference positions and stiffness values. The robustness of the proposed approach is further analyzed by applying a disturbance force on the actuator link. The results are compared with the linear method, showing that the proposed design can handle soft transitions in stiffness variation and ensure perfect tracking.

**Keywords:** compliant actuator; antagonistic variable stiffness actuator; identification; Hammerstein models; genetic algorithm

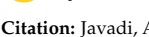



## 1. Introduction

In the realm of robotics, the limitations of traditional rigid robots have long been recognized as significant barriers to achieving fluid, adaptable motion akin to biological organisms. Rigid robots, while capable of precise movement and manipulation, often struggle in dynamic and unstructured environments due to their fixed stiffness and lack of compliance. This inherent rigidity results in inefficiencies, limited versatility, and safety concerns, particularly in applications requiring interaction with delicate objects or humans. Two complementary strategies are available: impedance control [1] and variable physical properties [2]. Variable stiffness actuators (VSAs) offer a transformative solution to these shortcomings using variable physical properties. By integrating compliant elements that can modulate stiffness levels dynamically, VSAs emulate the adaptability and versatility of biological systems. This flexibility enables robots equipped with VSAs to seamlessly adjust their stiffness according to task requirements, enhancing performance, energy efficiency, and safety. Moreover, the inherent compliance of VSAs facilitates smoother interactions with the environment, enabling robots to navigate complex terrains and collaborate with humans more effectively. These features are specifically important in some applications such as exoskeletons and rehabilitation [3,4]. Thus, the adoption of variable stiffness actuators represents a promising avenue for overcoming the limitations of rigid robots and

advancing the capabilities of robotic systems across a myriad of domains [5].

In the pursuit of developing advanced robotic systems, researchers have explored various variable stiffness actuator technologies, each presenting unique advantages and challenges. Series elastic actuators (SEAs) and parallel elastic actuators (PEAs) have emerged as prominent solutions, albeit with inherent disadvantages. SEAs, characterized by compliant elements such as springs integrated in series with actuators, offer notable advantages in precise force control and shock absorption [6,7]. However, their reliance on mechanical springs introduces inertia and limits their bandwidth, constraining their suitability for applications demanding rapid and dynamic movements especially with low stiffness. Similarly, PEAs, which incorporate compliant elements in parallel with actuators, excel in tasks requiring agility and responsiveness due to their enhanced bandwidth [8,9]. Because an elastic element with fixed stiffness is used in SEA and PEA, the link stiffness is fixed, limiting their usage in human-friendly robotic systems. These limitations underscore the need for alternative approaches. Serial variable stiffness actuators (SVSAs) [10–12] and antagonistic variable stiffness actuators (AVSAs) [13,14] provide alternative solutions to tackle these limitations. Antagonistic variable stiffness actuators offer a compelling solution by leveraging pairs of actuators with opposing stiffness characteristics. This design enables seamless stiffness modulation and dynamic adaptation to varying task requirements. Furthermore, AVSAs inherently enhance energy efficiency and stability, facilitating safer interaction with the environment and human operators. Thus, in contrast with the limitations posed by SEAs and PEAs, AVSAs present a promising avenue for advancing the capabilities of robotic systems, particularly in domains necessitating flexibility, adaptability, and energy efficiency [15].

Different AVSA technologies have been introduced, encompassing various actuation types, including the following: FAVSAs: fluidic AVSAs utilize hydraulic or pneumatic systems to achieve variable stiffness through the modulation of fluid pressure [16,17]. EMAVSAs: EMAVSAs employ electric motors or electromechanical components to adjust stiffness levels through mechanical means [18,19]. SMA AVSAs utilize shape memory alloys, which change shape in response to temperature variations, to achieve variable stiffness [20–22]. Cable-driven antagonistic variable stiffness actuators: cable-driven AVSAs utilize cables and pulleys to achieve variable stiffness by adjusting tension levels [23]. Magnetic antagonistic variable stiffness actuators: magnetic AVSAs utilize magnetic fields to modulate stiffness through the alignment of magnetic particles or materials [24].

In terms of control system design, several approaches have been investigated on AVSAs. In [25], two PID controllers were used for simultaneous position and stiffness adjustments. Meanwhile, a new method was introduced in [26] to transform an SEA to VSA using structured $H_\infty$ control. Moreover, the feedback linearization method has been used for position and stiffness control of tendon-driven VSAs [27,28]. Robust adaptive control of AVSAs was investigated in [13] in the presence of parameter uncertainties and external disturbances. For a bioinspired robot actuator, a cascaded control of AVSA was introduced in [29]. For an inflatable soft robot, the performances of the MPC and SMC approaches were compared in [16]. A linear extended state observer with estimation error compensation was applied to AVSA for the robust tracking control of position and stiffness [30]. The decoupled nonlinear adaptive control of position and stiffness for a pneumatic soft robot with a McKibben muscle was proposed in [31]. In [32], a nonlinear model predictive control approach was applied to a variable stiffness actuator.

Most existing results used an established theoretical model of the whole system (or some subsystems) to control the position and stiffness (model-based design). In other words, they derive a model for the position and the stiffness based on the motors' positions, and a static relationship between the motors' positions on one side and the position and stiffness of the link on the other side is obtained. These static relations can be inversely used to obtain desired motor positions. The model-based design performance is directly influenced by the parameter estimation accuracy. The model parameters include inertia, damping, and stiffness coefficients of the motors and the link. If the motor dynamic is

considered in the model, then the motor parameters must also be estimated. Alternatively, the dynamic of the system can be identified as a black box (data-driven method) and used for controller design. Data-driven control of VSAs using the identification method applied to the subsystems of VSA is reported in very few works in the literature [29,33]. The model-based design has several advantages, including the following:

- Modeled subsystems can be used for a local controller design (cascaded design [29]).
- Unmeasurable variables (e.g., stiffness) can be theoretically modeled; hence, the model-based control design is possible for them.

  Meanwhile, the model-based design has some disadvantages:

- The control performance depends on the parameter estimation accuracy; that is, the parameters should be measured or estimated accurately enough to have less model uncertainty.
- Many undesired effects are not considered in the theoretical model, such as the dead zone, backlash, and friction.
- Some of the plant subsystems may have unknown behaviors or are hard to model theoretically.

Because stiffness is not a measurable variable, several methods are used to obtain it, such as estimation [34] and theoretical modeling [16,24,35]. The theoretical modeling of stiffness has the same aforementioned disadvantages. Meanwhile, estimating stiffness requires a theoretical model of the system [34]; hence, the same disadvantages also apply to this case. To resolve the disadvantages of a model-based design, the identification approach as a data-driven method can be used alternatively. In this case, there is no need for model parameter measurement. In addition, all unknown effects, such as backlash, friction, and dead zone, are included in the identified model. The main benefit of the identification approach is that all unknown dynamics and subsystems of the VSA can be covered in the resulting model. Enough sensors (like encoders) in the right place enable the identification method to obtain a model even for the subsystems.

### 1.1. Related Works

Existing works related to the identification approach only extracted a model for the subsystems of the entire VSA; that is, they utilized a combination of theoretical modeling and identification methods. For example, in [29], an identified model was obtained from input voltage to motor positions by applying PRBS signals in a closed-loop configuration. However, the nonlinear relationship among motor positions and the link position and stiffness were obtained theoretically based on the physical VSA configuration (the so-called static decoupling method). The main issue with this (and fully theoretical modeling) approach is that any asymmetry between two motor behaviors or two nonlinear springs (or any other nonlinear elastic elements) and any different initial pretension result in some errors in position and stiffness tracking. This is because only the motor positions are controlled in the closed-loop configuration using position encoders, and from motor positions to the link position or stiffness, no sensor is involved and prone to tracking error.

A neural network is used in [33] on a feed-forward path to control a Qbmove marker pro, which is a variable stiffness actuator. The neural network in [33] only emulates the static equilibrium of the system, and the design is not fully data driven. More specifically, the neural network tries to produce two motors' positions based on the given output position and stiffness. This means that the NN is only emulating system behavior from motors' positions (not the voltages that are real inputs of the system) to the output position and stiffness. In other words, the motors are supposed to be ideal, and their dynamics are not included inside the NN. However, in our proposed method, we have supposed the whole system from input voltage to the output position and stiffness as a black box and identified it by some linear models.

To address the above-mentioned issues, we suggest the use of the identification method from input voltages of the motors to the link position (instead of motor positions). In this

way, the position control accuracy will be equal to the link position encoder accuracy, whose precision is certainly more than that of the open-loop decoupling approach. Unfortunately, stiffness is not a measurable quantity, and the identification method is thus not implementable. Instead of theoretical modeling for stiffness, a novel stiffness production approach is thus suggested. Experimental observations showed that when the difference between two input voltages of the motors is kept constant, the stiffness of the link is approximately equal at any position. However, the stiffness value in the CW and CCW directions is insignificantly distinctive because of the asymmetry in the motor dynamics and other hardware components. By applying different predefined voltage differences and measuring the output stiffness using a load cell sensor, a look-up table can be obtained between the voltage difference and the link stiffness, which can be approximated by a polynomial function. Using the fitted polynomial, the desired stiffness can be obtained by applying the obtained voltage difference of the motors.

### 1.2. Contributions

In this paper, our primary objective is to enhance the performance of the closed-loop system by employing alternative methods for system modeling. For the first time, we apply a data-driven approach to an AVSA without necessitating any theoretical modeling prerequisites. Previous attempts have relied on a combination of theoretical methods and identification approaches [29,33]. In addition, this is the first work using the voltage difference method for stiffness production. Furthermore, we have established a nonlinear elastic structure using a simple linear spring. Different mechanisms have been exploited to produce nonlinear elasticity, such as nonlinear spring [29], inflatable joints [16], twisted string actuators [27], mechanical-rotary variable impedance actuator [18], and twisted-coil polymer [36]. The proposed elastic mechanism is a simple structure including a usual linear spring. By changing the stiffness of this linear spring and the dimensions of the elastic elements, the nonlinear force–deflection curve and, hence, the stiffness of the elastic structure can be adjusted. In addition, the suggested voltage difference method can adjust the stiffness quite fast. The stiffness change speed is directly related to the motors' dynamics, and the time constant of the motors determines the stiffness adjustment speed. Unlike existing methods for stiffness adjustment based on an accurate change of motors' positions, in the voltage difference method, the voltage difference drives the load in the intended stiffness value regardless of the position, pretension of the springs, and initial condition of the system.

Moreover, the proposed design only requires the measurements of link position (output-feedback). In many existing works, positions and velocities of motors and a link are required for a control system (state-feedback). Such output-feedback control system requires less measurement sensors in the hardware and simplifies the control system. In the case of model-based designs, two approaches are possible: measuring all states by sensors and estimating unmeasured states by some observers. If all states are intended to be measured, a large number of sensors will be required, which will result in an expensive hardware with many sensors. On the other hand, if unmeasured states are intended to be estimated by observers, the control system will be more complicated. The proposed output-feedback controller reduces the overall control system complexity and, at the same time, requires a low number of sensors (only one encoder for a link position).

The methods for acquiring models and designing PID controllers are well established and widely recognized. The innovation of the suggested design lies in its distinctive utilization of these strategies. Through the utilization of the voltage difference method to induce stiffness and framing the control of the MIMO AVSA system as a SISO system with different stiffness levels, the complexity of system dynamics can be reduced. This facilitates a more efficient application of existing identification and control system design methodologies.

Identification-based methods applied to variable stiffness actuators typically aim to identify a linear model for the underlying system [29]. However, linear models may fail to capture nonlinear characteristics of the system, such as nonlinear spring behavior, friction,

and dead zones. Given the prevalence of friction effects in most geared motors, a nonlinear model such as the Hammerstein model is expected to provide a better fit to the measurements, as friction can be approximated by a polynomial static function, a key feature of the Hammerstein model [37]. This paper utilizes nonlinear Hammerstein models to describe the position dynamics of the AVSA setup. Two controllers are introduced for the AVSA system. In the first method, five different PID controllers are presented for each stiffness level. A corresponding PID controller is selected based on the stiffness level (voltage difference). Switching among different controllers avoids a soft transition of stiffness and position. To resolve this issue, a set of optimal PID gains with linear interpolation is proposed for a single PID controller, eliminating the need for switching and enabling smooth stiffness and position control.

The main contributions of this paper can be summarized as follows:

- The stiffness of the output link is created by applying a voltage difference between the input voltages of the motors.
- Nonlinear Hammerstein models are employed to model the position dynamics, effectively capturing nonlinearities inherent in the AVSA setup, including gearbox friction.
- PID gains are optimized using a genetic algorithm for predetermined stiffness and position values.
- Utilizing a single PID controller with interpolated optimal gains enables seamless control over both position and stiffness, facilitating smooth transitions in stiffness and position.

The rest of the paper is organized as follows: Section 2 provides a comprehensive overview of the hardware setup, detailing the nonlinear elastic elements incorporated. Within Section 3, the initial focus lies in presenting the stiffness model, followed by the identification of position dynamics using nonlinear Hammerstein models. Moreover, this section entails a comparative analysis between the fitness of nonlinear Hammerstein models and their linear counterparts. Section 4 delves into the design and optimization of PID control, including the utilization of genetic algorithms to fine-tune the gains. Sections 3 and 4 include the research methodology of the paper by analyzing and comparing different methods of identification and control design methods. Experimental findings and a comparative assessment with an existing method are expounded upon in Section 5. Subsequently, Section 6 presents significant insights and observations pertaining to the proposed design. Finally, Section 7 encapsulates the paper with concluding remarks.

## 2. Hardware Overview

Different actuation components have been utilized in previous AVSA systems, including an equivalent quadratic torsion spring [38], twisted strings [27], compliant air bladders [16], McKibben muscles [39], tendon-driven actuators [13], and a spiral disk regulating mechanism [40]. The present configuration employs a wired tendon-driven antagonistic setup. This configuration has been widely used in the literature and is an adequate implementation for testing the proposed stiffness production approach and the overall control system. Subsequently, the upcoming section will provide a breakdown of the manufactured AVSA setup, encompassing both mechanical and electrical elements.

### 2.1. Setup Specifications

A schematic view of the AVSA configuration is shown in Figure 1a. Two 60 rpm 12 V DC motors with maximum torques of 20 Nm were connected to the load pulley through two nonlinear springs. Our expected speed at the output link was 30 rpm, which can be provided by the selected 60 rpm motor. The diameters of the pulleys for motors 1 and 2 and the load were all 8 cm, and the link (load) length was 9.175 cm. A load cell was mounted below the link pulley to measure the output torque (force). The load cell maximum measurable load was 200 N. The motors, link, and load cell had their own encoders (1024 pulse/rev). The encoder measurements were sent to MATLAB Simulink

via a hardware-in-the-loop (HIL) interface board. After data processing, the produced commands to the motors were returned through the same HIL channel.

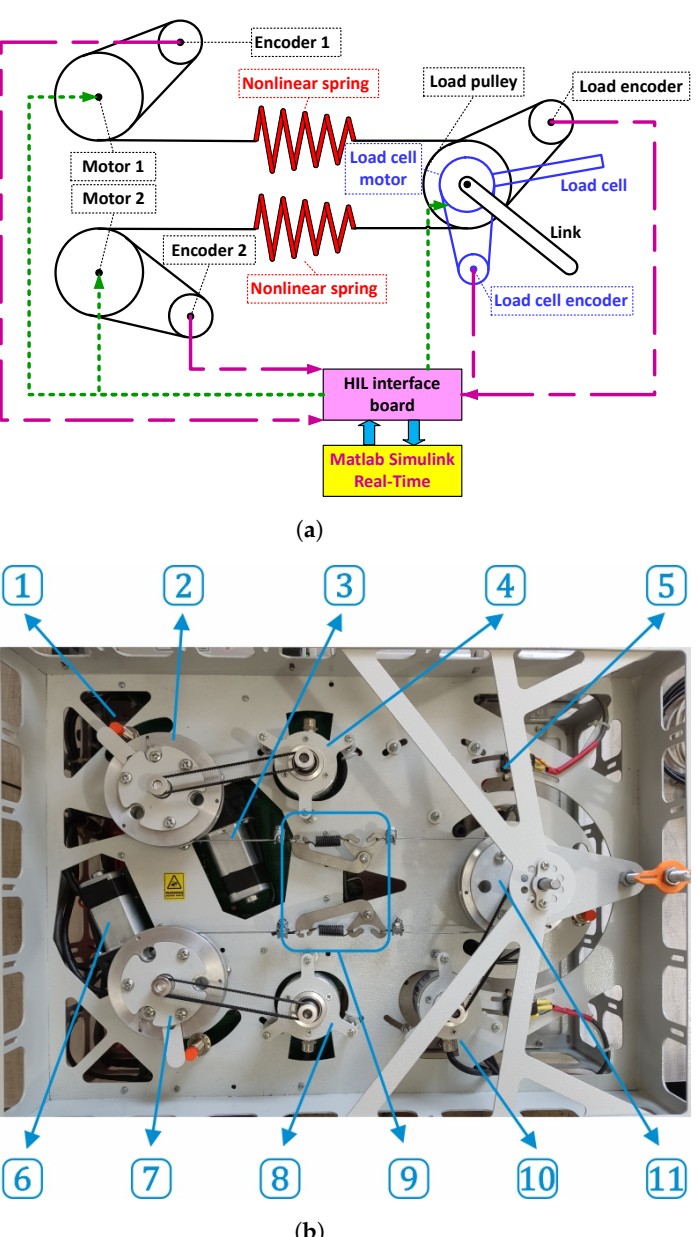

(**a**)

(**b**)

**Figure 1.** (**a**) Block diagram of the tendon-driven AVSA. Magenta and green colors represent the encoder readings and control commands sent to the motors, respectively. (**b**) Front view of the hardware configuration: (1) proximity sensor, (2) pulley of motor 1, (3) motor 1, (4) encoder 1, (5) limit switch, (6) motor 2, (7) pulley of motor 2, (8) encoder 2, (9) nonlinear springs, (10) load encoder, and (11) pulley of the output link.

The front view of the hardware implementation for AVSA is depicted in Figure 1b. Motors 1 and 2 and the output link were connected to their corresponding encoders using timing belts. The output link could freely rotate from $-90°$ to $+90°$. The mounted proximity sensors could be used to align the pulleys in certain positions and were useful for the homing scenario of the setup. Two limit switches were placed at two endpoints of the working region to avoid any damage to the hardware by turning off the motor power. An M8 screw was mounted at the ending point of the output link, which continued down to reach the load cell at the bottom of the hardware.

The back side of the setup is illustrated in Figure 2. Two AT91SAM3X8E Arduino Due processors with a clock speed of 84 MHz were utilized in the hardware. One of which was responsible for the HIL connection between the hardware setup and MATLAB Simulink in real time. The sample time for data transfer (receiving measurements and transmitting commands) was 12 ms. The other processor managed the interface board and the safety units. Several protection fuses were exploited to turn off the motors and board electricity when the motor currents crossed a predefined threshold (8 A). The load cell and the output link were mounted in a concentric configuration. The load cell motor rotated the load cell bar until hitting the M8 screw (which was fixed to the output link) and made a deflection at the link. The resulting deflection could be measured from the link encoder, and the applied torque can be measured from the load cell sensor. With this mechanism, the output stiffness can be calculated.

The existing configuration comprises two motors utilized for actuation and one motor designated for stiffness estimation. Furthermore, each motor is equipped with its own position sensor (encoder) for measuring the position, while torque is measured using a load cell sensor. Consequently, the entirety of the setup can be conceptualized as a network of sensors and actuators.

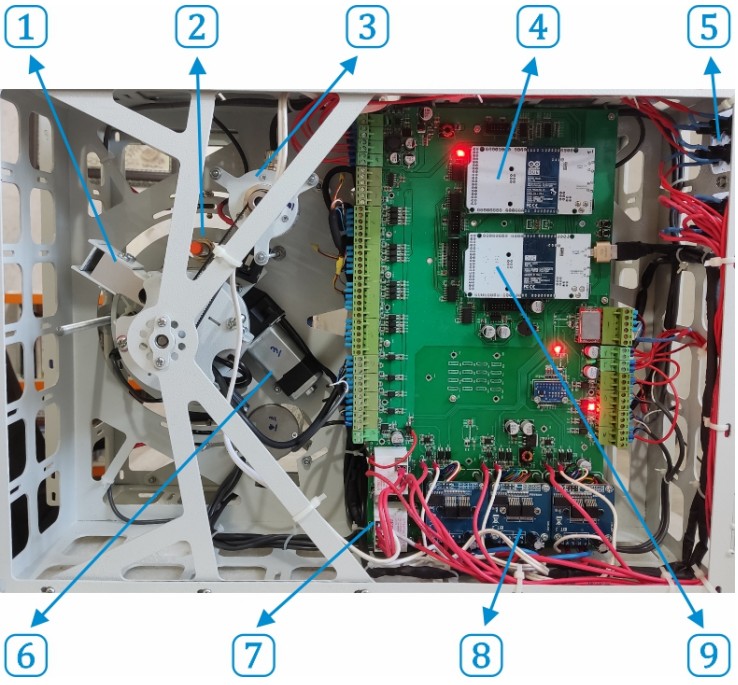

**Figure 2.** Back view of the hardware configuration: (1) load cell sensor, (2) proximity sensor of the load cell, (3) encoder of the load cell motor, (4) HIL processor, (5) protection fuses for motors and processors, (6) load cell motor, (7) safety relay unit, (8) DC motor drivers, and (9) main processor.

*2.2. Nonlinear Spring Mechanism*

It is well known that the torque–deflection curve of the spring in AVSA should have nonlinear behavior to have variable stiffness in the output link [15]. Here, the nonlinear behavior was provided by the mechanism shown in Figure 3b. The dimensions for the steel plates are shown in Figure 3a. The diameter of the linear spring is 10 mm, and the wire thickness is 1 mm. All plate thicknesses are equal to 1 mm. The proposed structure has nonlinear behavior, since by increasing the external force at two ends of the mechanism, the perpendicular component of the force on the first wing of the smaller plate is reduced. This reduction was reflected on the other wing (which was attached to the spring). Hence, the effective pulling force on the spring was reduced, which led to a smaller deflection. A larger force with less deflection represented higher stiffness values. This claim was proved by measured data in the following.

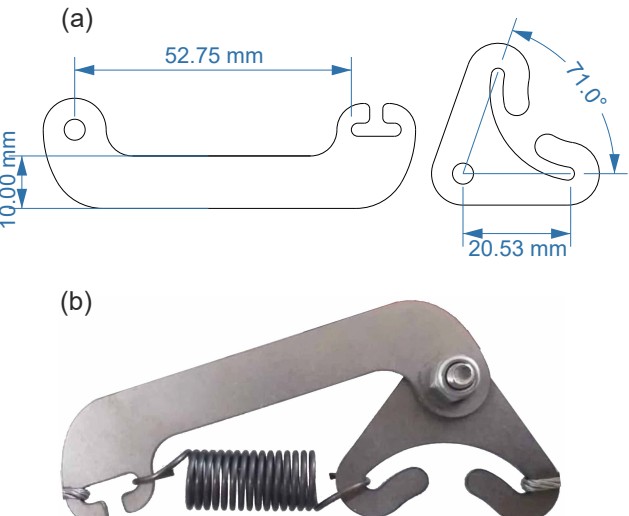

**Figure 3.** Nonlinear spring mechanism composed of two metal plates and a simple linear spring:
(**a**) steel plate dimensions and (**b**) prototype of a manufactured nonlinear spring.

Using one of the DC motors and the load cell readings, the nonlinear characteristic of
the nonlinear spring can be obtained. The resulting force–deflection curve is depicted in
Figure 4. A third-order polynomial was fitted to the measured data using MATLAB's curve
fitter tool and is drawn in Figure 4, which confirmed the nonlinear behavior of the spring
mechanism. The fitted polynomial is as follows:

$$F_s = 3.906 \times 10^6 x^3 - 1.578 \times 10^4 x^2 + 210.2x + 0.4221 \tag{1}$$

where $F_s$ is the applied force and $x$ is the corresponding deflection of the mechanism.

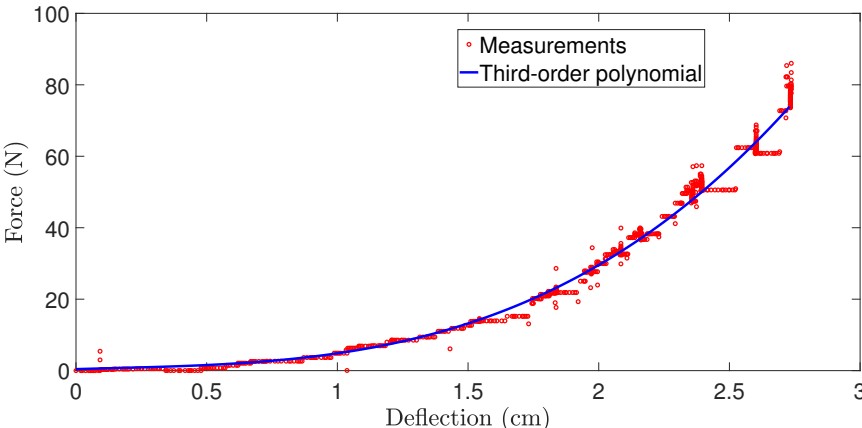

**Figure 4.** Force–deflection characteristics of the nonlinear spring mechanism.

This setup has been extensively employed in the literature and serves as a suitable
implementation for evaluating proposed approaches to stiffness production and the overall
control system.

**Remark 1.** *Concerning the hardware, all components of the presented AVSA setup are crucial for
accurate functioning in associated experiments. Notably, some components are specifically required
for particular experiments. For instance, the load cell and the connected motor are utilized solely for
stiffness measurement to establish the stiffness–voltage difference relationship through polynomial
fitting. In regular operational conditions, the load cell and connected motor remain inactive. Another
instance involves the connection of two encoders to the two motors; however, their measurements
are not utilized in typical working conditions and only the link position measurement is fed back.*

## 3. Identification of AVSA

As discussed earlier, in the proposed approach, the stiffness is produced by applying different voltages to the DC motors. In addition, the dynamic model of the position is obtained by applying PRBS signals to the input voltages, and the resulting link positions together with the input voltage are consequently used to identify a model. It is presumed that during the time interval when PRBS signals are imposed on the system, the position of the link remains within the permissible endpoints of $[-90°, 90°]$.

### 3.1. Stiffness Estimation

For stiffness adjustment, voltages were applied to the motors such that motors 1 and 2 rotated in CW and CCW directions, respectively. In this way, both nonlinear springs were pulled back, and the stiffness increased by increasing the voltage. As the DC motors were equipped with worm reduction gearboxes, the external force could not rotate the motor shaft, and only the motor could make a rotation on the gearbox shaft. Because of this property, when a torque was applied from one motor to the tendon (and hence the shaft of the other motor's gearbox), the other motor maintained the last position unless a voltage was applied to the second motor. $V_1$ and $V_2$ were assumed as the voltages applied to motors 1 and 2, respectively. If $V_1 - V_2$ was set to a known constant, then this voltage difference generates a deflection in the nonlinear springs and consequently produces a constant stiffness at the output. If this voltage difference was increased, the stiffness at the output increased as a result. Therefore, by applying different voltage differences, the desired stiffness at the output link could be adjusted. Remarkably, two different sets of voltages with the same difference produced the same stiffness, but the resulting positions were different. As an example, $V_1 = 4$ V, $V_2 = 1$ V and $V_1 = 5$ V , $V_2 = 2$ V led to the same stiffness; however, in the second case, the rotation speed was higher, and hence, the positions were different. In addition, the shaft of the gearboxes does not start rotating below 2.7 V due to the presence of friction in the gearboxes. This threshold for input voltages introduces a dead zone to the actuation unit. In the following, some techniques will be employed to tackle these imperfections.

The same experiment can be repeated for the $V_2 - V_1$ case. Unfortunately, the two profiles were not the same because two motors and gearboxes were not exactly the same and unknown effects like friction and dead zone existed in the hardware (hysteresis effect). However, two different profiles can be obtained for the output link's CW and CCW directions. For estimating link stiffness, eleven equally spaced intervals of input voltage differences were applied to the DC motors starting from 0.5 V to 5.5 V at 0.5 V steps. The resulting curves are illustrated in Figure 5. The stiffness was computed using the following formula:

$$S = \frac{\partial \tau}{\partial \theta} \approx \frac{\Delta \tau}{\Delta \theta} \tag{2}$$

where $S$ is the link stiffness, $\tau$ is the link torque, and $\theta$ is the link deflection. To compute the stiffness, partial differentiation was approximated by a small variation of the torque ($\Delta \tau$) and the position ($\Delta \theta$). In the experiments, the deflection was kept as small as possible, mostly around $3°$, to improve the approximation in (2). The chosen encoders had a pulse count of 1024 per revolution, resulting in a resolution of $360/1024 = 0.3516°$. Consequently, a deflection of $3°$ can be attained within the encoder resolution. While it is feasible to increase the deflection value for greater precision in measurements, doing so may compromise the accuracy of the stiffness derivation outlined in Equation (2). Furthermore, the presence of a robust hardware framework, as depicted in Figure 1b, precludes the occurrence of structural deformation. This results from employing metal plates with a thickness of 2 mm during the fabrication of the frame, which encases the components.

The torque variation was obtained using load cell measurement as follows:

$$\Delta \tau = r \Delta F_{LC} \tag{3}$$

where $r = 9.175$ cm is the link length and $\Delta F_{LC}$ is the force variation measured by the load cell.

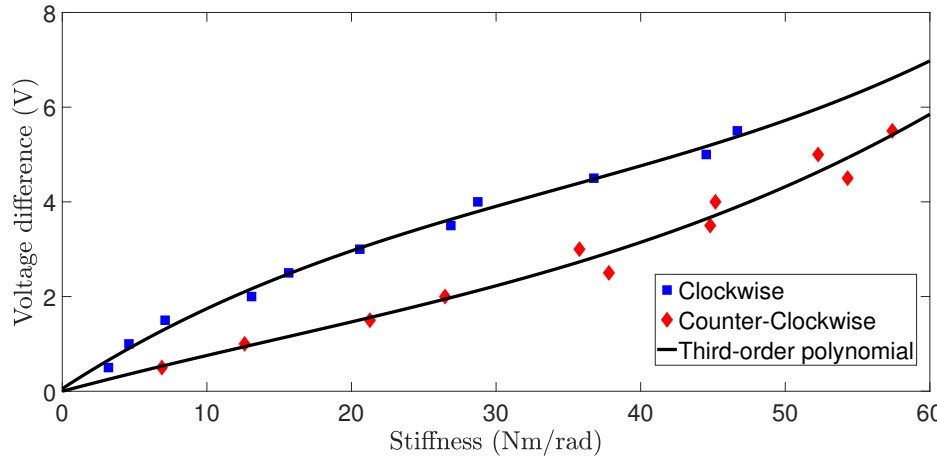

**Figure 5.** Link stiffness to voltage difference curve for the CW and CCW directions together with their third-order polynomial approximation.

As can be observed from Figure 5, link stiffness with respect to the same voltage difference has a different behavior with respect to the CW and CCW rotation directions of the load pulley. To obtain the required voltage differences for a certain desired stiffness, two third-order polynomials were fitted to the measured data by means of MATLAB's curve fitter tool. The stiffness-to-voltage difference formula for the CW and CCW directions was obtained as follows:

$$\Delta V_{CW} = 1.682 \times 10^{-5} S^3 - 7.392 \times 10^{-4} S^2 + 0.08132 S - 4.416 \times 10^{-4}$$
$$\Delta V_{CCW} = 3.199 \times 10^{-5} S^3 - 3.313 \times 10^{-3} S^2 + 0.199 S + 0.05393$$

(4)

where $\Delta V_{CW}$ and $\Delta V_{CCW}$ denote the voltage differences for the CW and CCW directions, respectively. Once the desired stiffness was selected, with respect to the sign of the desired position (CW or CCW), the corresponding formula was used from (4) to obtain the related voltage difference.

### 3.2. Linear Models

Stiffness characteristics were obtained for different voltage differences in the last subsection. Because the dynamics from the input voltage to the output position were dissimilar for different voltage differences, we obtained several linear models around some operating points. Five voltage differences were chosen in the interval $[0, 6 \text{ V}]$, namely, $|\Delta V| = 1, 2, 3, 4, 5$ V. $|\Delta V| = 1$ and $|\Delta V| = 5$ V generated the lowest and highest stiffnesses, respectively. Notably, when the input voltage was positive (CCW direction for the link), $\Delta V = V_1 - V_2$ was applied. Conversely, when the input voltage was negative (CW direction for the link), $\Delta V = V_2 - V_1$ was applied. In the first case, motor 1 pulled the tendon (and hence the second motor) with respect to the voltage difference and vice versa. Once the voltage difference was set, one of the voltages could be obtained from the other one. With this definition, only one input voltage was applied (instead of two voltages for two motors) for any given voltage difference. Now, we have a SISO system with the input voltage as the input and the link position as the output. For each of the five voltage differences $\Delta V$, PRBS signals were applied to the input voltage, and the link positions were logged. As an example, the logged data for $\Delta V = 3$ V are shown in Figure 6. One set of measurements was used for the identification step, and the other set was used for model validation.

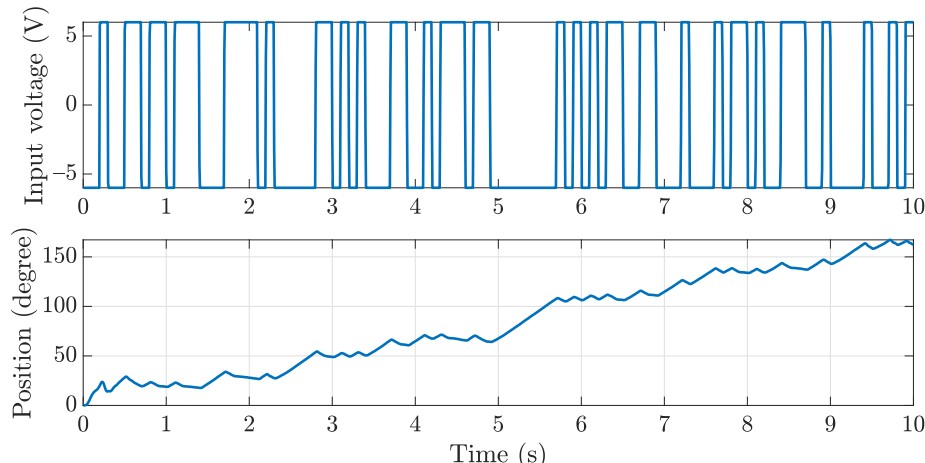

**Figure 6.** Measured data for voltage-position dynamic identification with $\Delta V = 3$ V. The upper plot shows the applied input voltage, and the lower plot illustrates the resulting link angle.

Using MATLAB's Identification toolbox, five linear models were fitted to the measured data as follows:

$$
\begin{aligned}
G_1(s) &= \frac{-1.123s^2 + 286.2s - 6.791 \times 10^4}{s^3 + 30.46s^2 + 5278s + 69.71} \\[4pt]
G_2(s) &= \frac{3.614s^2 - 789.1s - 2.896 \times 10^4}{s^3 + 131.7s^2 + 3298s - 197.7} \\[4pt]
G_3(s) &= \frac{-6.343s^2 - 1.501s - 0.9888}{s^3 + 0.3916s^2 + 0.024s + 9.344 \times 10^{-3}} \\[4pt]
G_4(s) &= \frac{-6.5s - 3.805}{s^2 + 0.1177s + 7.663 \times 10^{-4}} \\[4pt]
G_5(s) &= \frac{-5.696s - 2.66}{s^2 + 0.1241s + 3.707 \times 10^{-8}}
\end{aligned}
\tag{5}
$$

where $G_i(s)$ for $i = 1, 2, 3, 4, 5$ denotes the linear continuous transfer function from the input voltage to the link position for $|\Delta V| = i\, V$ and $s$ is the Laplace variable. According to MATLAB's Identification toolbox, the fitting level to validation data for the models $G_1(s)$ to $G_5(s)$ are 46.68%, 53.44%, 66.96%, 71.83%, and 92.93%, respectively. Interestingly, for higher voltage differences (higher stiffnesses), the model order was lower, which agrees with intuition. When the stiffness was high ($G_5(s)$ in (5)), the spring was stretched tightly and behaved like a rigid tendon; hence, the model order was reduced to two (the last term of the denominator is approximately zero). As the voltage difference was reduced and the stiffness became lower, the spring was not highly rigid and kept its elasticity; therefore, the model order for the lowest stiffness ($G_1(s)$ in (5)) increased to three. It is well known that the dynamic model of a DC motor from the input voltage to the speed has order 2 and that from the voltage to the angle has order 3. This extra pole is the integrator because the motor angle is the integral of the speed. The model with the highest stiffness $G_5(s)$ had an integrator that agreed with the theoretical model order of the DC motor without a spring mechanism.

**Remark 2.** *It is crucial to understand that the identified models are employed to characterize the voltage-to-position dynamics of the system rather than the stiffness behavior. As mentioned earlier, the stiffness at the output link is generated by the voltage difference of two motors through a fitted polynomial. Consequently, the control performance for link stiffness is not contingent on the precision of the identified position models. Instead, it is influenced by factors such as the accuracy of polynomial fitting, the precision of the load cell sensor, and the number of selected voltage differences for stiffness measurements.*

### 3.3. Nonlinear Hammerstein Models

In previous subsection, linear models have been identified for each stiffness level. Linear models might not capture the entire behavior of the AVSA system due to some factors such as friction and dead zone. Consequently, Hammerstein models are used to obtain better models of the system. A Hammerstein model is composed of two parts: a nonlinear static function, followed by a linear dynamic model [37]. A schematic diagram of the Hammerstein model is shown in Figure 7.



**Figure 7.** Schematic view of the Hammerstein model.

The nonlinear static function is supposed to be a polynomial of order $p$:

$$x^*(t) = r_0 + r_1 u(t) + r_2 u^2(t) + \cdots + r_p u^p(t) \tag{6}$$

Note that $x^*(t)$ is not a physical variable and cannot be measured. The linear dynamic system can be represented by the following transfer function:

$$
\begin{aligned}
G(q^{-1}) &= \frac{Y(q^{-1})}{X^*(q^{-1})} = \frac{B^*(q^{-1})}{A(q^{-1})} q^{-d} \\
A(q^{-1}) &= 1 + a_1 q^{-1} + a_2 q^{-1} + \cdots + a_{m_a} q^{-1} \\
B^*(q^{-1}) &= b_1^* q^{-1} + b_2^* q^{-1} + \cdots + b_{m_b}^* q^{-1}
\end{aligned}
\tag{7}
$$

where $Y(q^{-1})$ and $X^*(q^{-1})$ are $z$-transform of $y(t)$ and $x^*(t)$, respectively. $q^{-1}$ denotes a shift operator (unit delay), and $d$ represents the transport delay of the system. $m_a$ and $m_b$ indicate the number of poles and zeros of the linear dynamic model, and $G(q^{-1})$ is the $z$-transfer function of the linear part. Note that $b_0^*$ is supposed to be zero in the numerator of the transfer function due to the biproper nature of the AVSA system (i.e., the system output does not instantaneously follow the input voltage change). By cross multiplication of the transfer function in (7) and obtaining the inverse $z$-transform, we obtain the following:

$$y(t) + \sum_{i=1}^{m_a} a_i y(t-i) = \sum_{i=1}^{m_b} b_i^* x^*(t-d-i) \tag{8}$$

Substituting polynomial function (6) in (8) yields the following:

$$y(t) = -\sum_{i=1}^{m_a} a_i y(t-i) + \sum_{i=1}^{m_b} b_i^* \left[ r_0 + \sum_{j=1}^{p} r_j u^j(t-d-i) \right] \tag{9}$$

The difference Equation (9) represents the Hammerstein model of the system in the time domain. A thorough examination of recorded data (such as the measured data depicted in Figure 6) indicates the absence of any transportation delay within the system ($d = 0$). It should be noted that any potential time delay within the system, if present, could be shorter than the sampling time of 12 ms and, therefore, might not be discernible in the measurements. In addition, due to the substantial symmetry of the motors and gearboxes (which implies the symmetry in the polynomial function), the constant parameter of the nonlinear function (6) is also assumed to be zero $r_0 = 0$. Furthermore, it is assumed that

$m_b = 1$, and without loss of generality, $b_i^* = 1$. Considering the assumptions above, the difference Equation (9) can be rewritten as follows:

$$
\begin{aligned}
y(t) = &-a_1 y(t-1) - a_1 y(t-2) - \cdots - a_{m_a} y(t - m_a) \\
&+ r_1 u(t-1) + r_2 u^2(t-1) + \cdots + r_p u^p(t-1)
\end{aligned}
\tag{10}
$$

Now, the measured values $y_o(t)$ are used instead of the model output $y(t)$, and estimated parameters are plugged into the model (10), resulting in the following:

$$
\begin{aligned}
y_o(t) = &-\hat{a}_1 y_o(t-1) - \hat{a}_1 y_o(t-1) - \cdots - \hat{a}_{m_a} y_o(t - m_a) \\
&+ \hat{r}_1 u^1(t-1) + \hat{r}_2 u^2(t-1) + \cdots + \hat{r}_p u^p(t-1) + e(t)
\end{aligned}
\tag{11}
$$

The residual $e(t)$ in (11) arises from utilizing the measured values $y_o(t)$ instead of the model output $y(t)$ and employing parameter estimates rather than true values. This error definition is linear in parameters owing to the assumptions made, and hence, the least square method can be used for estimation. The presence of the error variable $e(t)$ avoids the computation of the output $y_o(t)$; however, Equation (11) can be used to obtain a one-step-ahead prediction of the output $\hat{y}(t|t-1)$ based on the measurements available up to time $t-1$. Therefore, a one-step prediction of the output can be written as follows:

$$
\begin{aligned}
\hat{y}(t|t-1) = &-\hat{a}_1 y_o(t-1) - \hat{a}_2 y_o(t-1) - \cdots - \hat{a}_{m_a} y_o(t - m_a) \\
&+ \hat{r}_1 u^1(t-1) + \hat{r}_2 u^2(t-1) + \cdots + \hat{r}_p u^p(t-1) \\
= &\, \psi^T(t)\hat{\theta}
\end{aligned}
\tag{12}
$$

where the data vector $\psi^T(t)$ and the parameter vector $\hat{\theta}$ are defined by the following:

$$
\begin{aligned}
\psi^T(t) &= \begin{bmatrix} -y_o(t-1) & \cdots & -y_o(t - m_a) | u(t-1) & \cdots & u^p(t-1) \end{bmatrix} \\
\hat{\theta} &= \begin{bmatrix} \hat{a}_1 & \cdots & \hat{a}_{m_a} | \hat{r}_1 & \cdots & \hat{r}_{m_b} \end{bmatrix}
\end{aligned}
\tag{13}
$$

Note that a data vector contains available input and output measurements and a parameter vector contains the unknown model parameters to be estimated. Subtracting (12) from (11), the error equation is obtained as follows:

$$
e(t) = y_o(t) - \hat{y}(t|t-1)
\tag{14}
$$

Assume that the input–output measurements are available for times $t = 1, 2, \ldots, m_a + N$. In this case, the data vector $\psi^T(t)$ will be filled for the first time at time $t = m_a$. By substituting (12) into (14), the observation equation is derived as follows:

$$
y_o(t) = \psi^T(t)\hat{\theta} + e(t)
\tag{15}
$$

For the time steps $t = m_a + 1, \ldots, m_a + N$, a system of $N$ equations can be established from (15) in the following form:

$$
\begin{aligned}
\mathbf{Y}_o &= \Psi\hat{\theta} + \mathbf{E} \\
\Psi &= \begin{bmatrix}
-y_o(m_a) & \cdots & -y_o(1)|u(m_a) & \cdots & u^p(m_a) \\
-y_o(m_a+1) & \cdots & -y_o(2)|u(m_a+1) & \cdots & u^p(m_a+1) \\
\vdots & & \vdots & & \vdots \\
-y_o(m_a+N-1) & \cdots & -y_o(N)|u(m_a+N-1) & \cdots & u^p(m_a+N-1)
\end{bmatrix} \\
\mathbf{Y}_o &= \begin{bmatrix} y_o(m_a+1) \\ y_o(m_a+2) \\ \vdots \\ y_o(m_a+N) \end{bmatrix}, \mathbf{E} = \begin{bmatrix} e(m_a+1) \\ e(m_a+2) \\ \vdots \\ e(m_a+N) \end{bmatrix}
\end{aligned}
\tag{16}
$$

In order to determine $m_a + p$ parameters, the number of measurements must be at least $m_a + p$, i.e., $N > m_a + p$. Assume the sum of squared errors as a cost function:

$$V = \mathbf{E}^T \mathbf{E} = \left(\mathbf{Y}_o - \Psi \hat{\theta}\right)^T \left(\mathbf{Y}_o - \Psi \hat{\theta}\right) \tag{17}$$

To minimize the cost function (17), the first derivative must be equal to zero:

$$\frac{dV}{d\hat{\theta}} = -2\Psi^T \left(\mathbf{Y}_o - \Psi \hat{\theta}\right) = 0 \tag{18}$$

The solution of the minimization problem (18) (least squares problem) is given by the following:

$$\hat{\theta} = \left(\Psi^T \Psi\right)^{-1} \Psi^T \mathbf{Y}_o \tag{19}$$

Note that $\left(\Psi^T \Psi\right)^{-1} \Psi^T$ is a well-known Moore–Penrose inverse of the data matrix $\Psi$ in (16). The optimal parameters of the Hammerstein model are given by (19), which minimize the sum of square errors between the measured outputs and model predictions. However, the existence of such optimal parameters inverse of the matrix $\Psi^T \Psi$ is required. The size of the matrix $\Psi \in {}^{N \times (m_a + p)}$ grows for larger measurements, but regardless of the measurements' size $N$, the size of $\Psi^T \Psi \in {}^{(m_a + p) \times (m_a + p)}$ is fixed by the number of model parameters. For the matrix $\Psi^T \Psi$ to be nonsingular, it is necessary for the system to be sufficiently excited. A PRBS signal with a period length $N$ is persistently exciting of the order $m_a + p$, if $N = m_a + p + 1$ [37]. This condition is satisfied for all measurement experiments. As an example, for $\Delta V = 1$ V, the final time is 30 s, and hence, the number of measurements is $N = 30/0.012 = 2500$, which is much higher than the number of model parameters $m_a + p + 1$.

For different values of $m_a$ and $p$, the optimal parameters were obtained, and then the RMSE values of the errors between measurements and the Hammerstein model were calculated. $m_a = 3$ and $p = 3$ led to the minimum RMSE values. Table 1 shows the optimal parameters of the Hammerstein models for each voltage difference.

**Table 1.** Hammerstein model parameters for different voltage differences.

| Parameters | $\Delta V = 1$ V | $\Delta V = 2$ V | $\Delta V = 3$ V | $\Delta V = 4$ V | $\Delta V = 5$ V |
|---|---|---|---|---|---|
| $\hat{a}_1$ | −2.51 | −2.22 | −2.25 | −2.03 | −1.94 |
| $\hat{a}_2$ | 2.17 | 1.65 | 1.67 | 1.3 | 1.17 |
| $\hat{a}_3$ | −0.655 | −0.432 | −0.423 | −0.268 | −0.234 |
| $\hat{r}_1$ | 0.0122 | 0.0101 | 0.00934 | 0.00471 | 0.00497 |
| $\hat{r}_2$ | $4.58 \times 10^{-5}$ | $5.29 \times 10^{-4}$ | $5.06 \times 10^{-4}$ | $4.99 \times 10^{-4}$ | $6.37 \times 10^{-4}$ |
| $\hat{r}_3$ | −0.00147 | $-8.67 \times 10^{-4}$ | $-4.89 \times 10^{-4}$ | $-2.65 \times 10^{-4}$ | $-1.95 \times 10^{-4}$ |

The nonlinear static function of the Hammerstein model for $\Delta V = 1$ V is illustrated in Figure 8. As seen from Figure 8, the nonlinear static function has a negative sign, which was predictable from the negative sign of $\hat{r}_3$ in Table 1. This occurred due to the clockwise definition of the positive orientation of the position. It is remarkable that the output of the nonlinear function $x_s^*$ is close to zero for input voltages in the interval $[−2.7, 2.7]$, which is consistent with the dead zone of the motors.

To compare the obtained Hammerstein models, the RMSE of the position for the linear models and the Hammerstein models were compared with each other. Table 2 shows the RMSE of the errors between measurements and the Hammerstein models for different voltage differences. RMSE values clearly confirm the advantage of the Hammerstein models over the linear models. Overall, the improvement is about 90%. The enhancement primarily stems from integrating the nonlinear static function into the Hammerstein model, effectively capturing and modeling the dead-zone characteristics of the gearboxes.

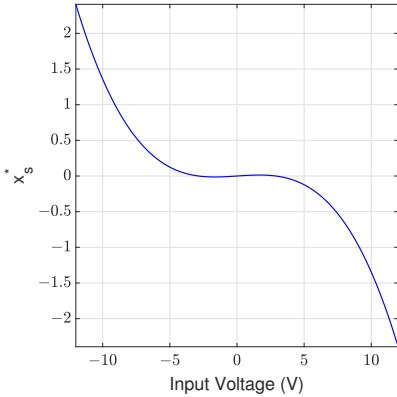

**Figure 8.** Nonlinear function of the Hammerstein model for $\Delta V = 1$ V.

**Table 2.** RMSEs for the proposed Hammerstein models and the linear models obtained from the least squares method.

| Voltage Difference (V) | Linear Model | Hammerstein Model | Improvement Percentage |
|---|---|---|---|
| $\Delta V = 1$ | 22.229 | 2.552 | 88.52 |
| $\Delta V = 2$ | 41.837 | 5.173 | 87.635 |
| $\Delta V = 3$ | 44.355 | 4.169 | 90.601 |
| $\Delta V = 4$ | 48.753 | 4.206 | 91.373 |
| $\Delta V = 5$ | 46.022 | 1.824 | 96.037 |

To illustrate the performance of the Hammerstein model, the predicted positions of the linear models and the Hammerstein models together with the measurements for $\Delta V = 1$ V are shown in Figure 9. Since the measurements were logged in degrees, the identified models' output is also in degrees. Obviously, the predicted output position from the Hammerstein model surpasses that of the linear model by a significant margin.

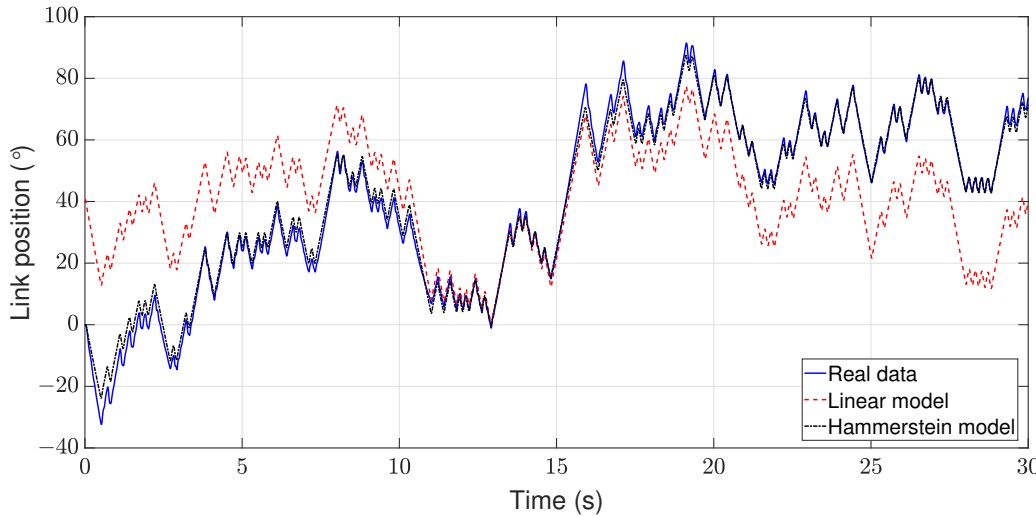

**Figure 9.** Position measurements, linear and Hammerstein model outputs for $\Delta V = 1$ V.

## 4. Control Design

The PID control method stands out as the prevailing solution for addressing real-world control challenges [41] due to its simple structure, easy implementation, and acceptable performance. In this section, two control approaches with a PID structure are introduced for linear and nonlinear Hammerstein models.

*4.1. Switching PID Control for Linear Models*

For any given model in (5), a PID controller was designed to have perfect tracking for the link position. The general form of the continuous PID controllers is as follows:

$$C_i(s) = P_i + \frac{I_i}{s} + D_i \frac{N_i}{1 + \frac{N_i}{s}} \tag{20}$$

where $P_i$, $I_i$, $D_i$, and $N_i$ for voltage differences $i = 1, 2, 3, 4, 5$ V are proportional, integral, derivative, and filter coefficients, respectively. The filter used in the derivative term was useful for reducing the measurement noise of the sensors and for suitable implementation. Table 3 shows that the derivative gains were quite small, and for $i = 4$ V, a PI controller was obtained. Because all controllers had integral terms, for the steplike desired positions, perfect tracking can be achieved.

**Table 3.** PID gains with respect to the voltage difference.

| $\Delta V$ | PID Coefficients | | | |
|---|---|---|---|---|
| i | $P_i$ | $I_i$ | $D_i$ | $N_i$ |
| 1 | −0.476 | −0.544 | 0.0115 | 10.42 |
| 2 | −0.421 | −0.421 | 0.013 | 7.5 |
| 3 | −0.907 | −0.961 | −0.025 | 21.75 |
| 4 | −0.905 | −0.772 | 0 | 0 |
| 5 | −0.96 | −0.673 | −0.04 | 8.77 |

Each PID controller in (20) was responsible for the corresponding voltage difference. To select the best PID controller for any given desired stiffness (with related voltage difference from (4)), the following condition is defined:

$$i = \begin{cases} 1 & 0 < \Delta V < 1.5 \\ 2 & 1.5 \le \Delta V < 2.5 \\ 3 & 2.5 \le \Delta V < 3.5 \\ 4 & 3.5 \le \Delta V < 4.5 \\ 5 & 4.5 \le \Delta V \end{cases} \tag{21}$$

Although the PID controller $C_1(s)$ was designed for $\Delta V = 1$ V, it can be used for the surrounding region $0 < \Delta V < 1.5$. For better performance, the voltage intervals can be made smaller. However, the number of PID controllers is increased, which complicates the design.

Soft Switching among Controllers

An active PID controller was selected with respect to the changing desired stiffness values using the rules in (21). Thus, switching among the five PID controllers was inevitable. This switching during system operation can produce jumps at the control input because the outputs of the five PID controllers are not the same at any given time. To resolve this issue, the following low-pass filter was placed after the PID blocks:

$$G_S = \frac{3}{s + 3} \tag{22}$$

Whenever a jump was intended to occur, the low-pass filter (22) was activated to switch softly from the previous PID to the next PID controller. Otherwise, the active PID continued to work in the HIL. Overall, the PID controllers designed with the switching rule in (21) along with the low-pass filter in (22) result in a smooth transition of stiffness values and achieve precise tracking for the step-shaped reference position.

A schematic diagram of the switching PID control system design is illustrated in Figure 10. All PID controllers have the same continuous structure in (20) with PID gains in

Table 3. Based on the voltage difference provided by (4), a corresponding PID controller is chosen from the switching rule in (21). The designed LPF (22) smooths the transition from one PID output to the other whenever switching occurs. The voltage difference and the voltages of the motors are obtained by (4). Link position is measured by the encoder and subtracted from a reference position to construct the tracking error. It is clear from Figure 10 that the proposed design is an output feedback control system without any current, velocity, or position control loop for individual motors.

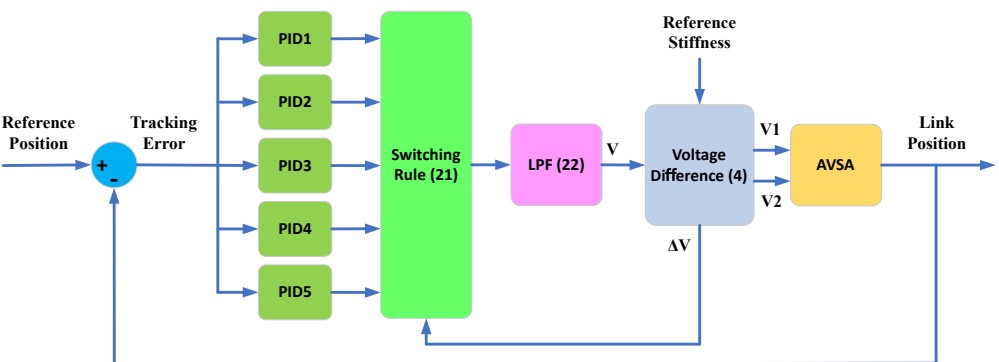

**Figure 10.** Schematic view of the closed-loop system.

### 4.2. PID Control for Hammerstein Models

A discrete PID controller is used to control the AVSA position dynamics since the obtained Hammerstein models are also discrete. The following implementation for PID is considered:

$$G_{PID}\left(q^{-1}\right) = P + \frac{IT_s}{2}\frac{1+q^{-1}}{1-q^{-1}} + \frac{DN}{1 + \frac{NT_s}{2}\frac{1+q^{-1}}{1-q^{-1}}} \tag{23}$$

where $P$, $I$, and $D$ are proportional, integral, and derivative gains, respectively. Instead of an ideal derivative that can amplify the measurement noise, a filtered derivative term is utilized with the parameter $N$. For very large values of $N$, this term approaches the ideal derivative. For discretizing integral and derivative terms, the trapezoidal method has been used as an implicit method. It is also second-order accurate and guarantees the stability of discretization in case the continuous PID is stable [42].

Selecting a PID controller is motivated by several factors. First, its inclusion of an integral term ensures precise tracking of steplike position references. Additionally, its straightforward implementation adds to its appeal.

Optimization with GA

There are numerous heuristic algorithms available for optimization purposes, such as GA, PSO, and SA. Each algorithm may excel in specific applications. When dealing with complex and highly nonlinear optimization problems, it is crucial to select the most suitable method based on factors like computational complexity, optimization time, and adjustable parameters. Literature comparisons indicate that for standard optimization tasks involving few variables (e.g., the four PID parameters in our design), the performance of these algorithms is comparable [43,44]. However, since our optimization process is conducted offline and the resulting gains are applied to the closed-loop system, considerations such as computational efficiency, convergence speed, and the number of adjustable parameters are of less significance in our design. The obtained results confirm that the gains computed by GA yield acceptable minimums for the cost function, constituting satisfactory suboptimal solutions, while other algorithms may primarily enhance computational efficiency or convergence speed. Consequently, the GA algorithm was used in the current design to find optimal PID gains.

GA is an evolutionary algorithm inspired from the biological development of species with mating selection and survival of the fittest [45]. The versatility of the genetic algorithm has resulted in its extensive application across various optimization problems. The brief and basic algorithm of GA is illustrated in Figure 11. The population containing underlying parameters is initialized at first. Crossover allows the combination of the genetic materials of two or more solutions. In a mutation step, the solution is disturbed by some random changes. A fitness function is defined representing the quality of the solutions. The fitness function is computed for the whole population. Based on predefined selection criteria, the best offspring solutions are selected to be parents in the new population. If the obtained population satisfies the desired fitness criteria, the optimal parameters are ready to use and the algorithm is terminated. Otherwise, the selected parents are used to repeat the algorithm from the crossover step.

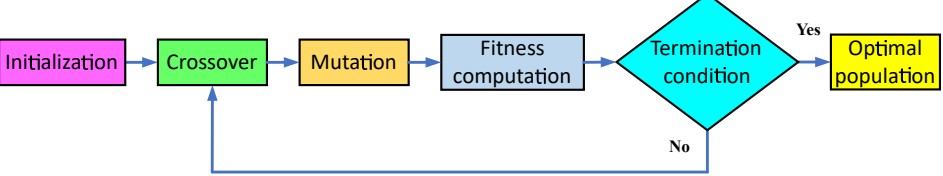

**Figure 11.** A schematic diagram of genetic algorithm steps.

In the current design scenario, the population of GA comprises four PID parameters, $P$, $I$, $D$, and $N$, which are subject to optimization. Due to the nonlinear nature of Hammerstein models, the optimized PID gains vary depending on the amplitude of the reference position $A_r$. Hence, it is necessary to choose various reference position values and run optimization for each amplitude level. Note that the sign of the reference position is not important in this regard, and only the absolute value of the amplitude is important. In other words, step signals with the same amplitudes but with opposite signs lead to the same optimized gains. We have selected amplitude levels ranging from 10° to 90°, in increments of 10°, resulting in nine levels. Additionally, there are five Hammerstein models corresponding to five stiffness levels. Thus, there exist 45 combinations of different levels for position amplitude and stiffness (Hammerstein models).

In the optimization process, a population size of 30 is assumed, and the termination criteria are set to a maximum of 100 iterations. In addition, the fitness function is supposed to be an RMSE of the position tracking error ($e_\theta = \theta_r - \theta$) passed through the following low-pass filter:

$$G_{LPF}\left(q^{-1}\right) = \frac{0.05q^{-1}}{1 - 0.95q^{-1}} \tag{24}$$

The inclusion of LPF (24) is essential because, without it, the optimized PID gains result in a very rapid response from the closed-loop system, leading to excessively high gains. These high gains, in turn, cause large control inputs that violate the maximum allowable voltages of the motors and result in poor control performance. To mitigate this issue, a smoother version of the reference step function can be achieved by filtering the reference input. A step signal with the amplitude $A_r$ is assumed as the reference position $\theta_r$.

The results of GA optimization for PID gains are depicted in Figure 12. The acquired 45 gains are denoted by red circles in the same figure. It is notable that all proportional ($P$), integral ($I$), and derivative ($D$) gains exhibit negative values. This phenomenon arises due to the negative nonlinear static function in all Hammerstein models, as detailed in Section 3.3. Examining the upper-left quadrant of Figure 12, it becomes evident that the absolute value of the proportional gain increases with $\Delta V$ (stiffness) for nearly all position amplitudes. Conversely, the integral gain demonstrates minimal variation with changes in stiffness; its primary sensitivity lies in the amplitude of the reference position. The derivative gain exhibits a higher amplitude with increased stiffness, yet its dependency on the reference position amplitude remains limited. Furthermore, the derivative filter

parameter $N$ exceeds 10 for all $\Delta V$ and $A_r$ values. This parameter has the least influence on the performance compared with the other parameters.

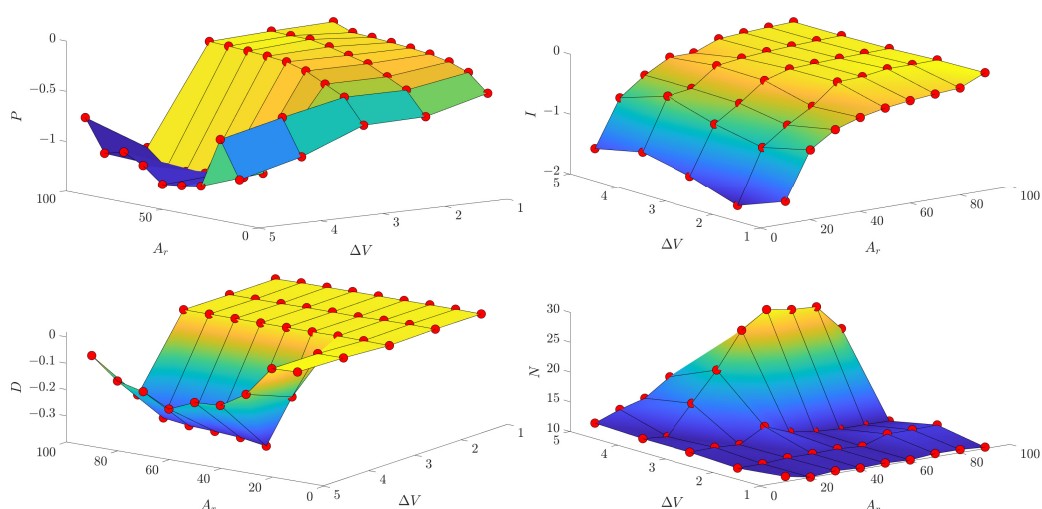

**Figure 12.** Proportional ($P$), integral ($I$), and derivative ($D$) gains together with derivative filter parameter ($N$) obtained from GA in terms of five voltage differences ($\Delta V$) and nine reference position amplitudes ($A_r$).

A schematic diagram of the closed-loop system is shown in Figure 13. Knowing $A_r$ and $\Delta V$, the $P$, $I$, $D$, and $N$ gains are calculated by linearly interpolating 45 points in Figure 12. Interpolation lines are shown with black color in Figure 12. Since the reference position and stiffness (voltage difference) can change abruptly, the following low-pass filter was used to smooth the gains for better performance:

$$G_{gain}(s) = \frac{1}{1 + 0.05s} \tag{25}$$

LPF (25) was discretized by the sample time of $T_s = 12$ ms for implementation. Once the reference stiffness and PID output are available, the voltage difference $\Delta V$ and motors' voltages can be obtained using (4).

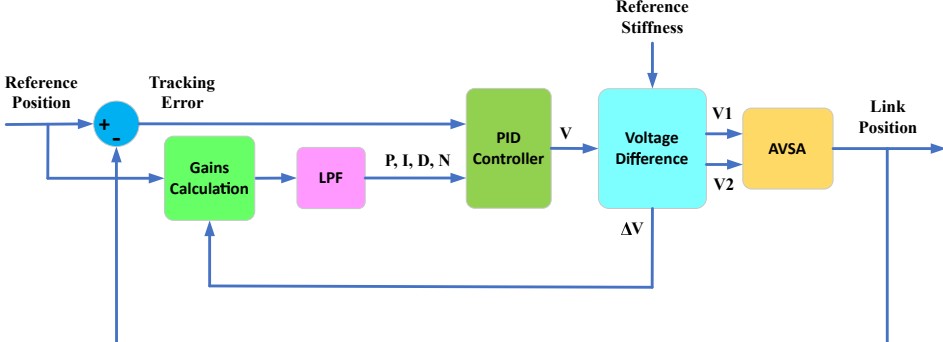

**Figure 13.** Closed-loop configuration of the control system.

## 5. Experimental Results

In this section, the performance of the proposed method will be evaluated. Several different scenarios were conducted, and a switching PID controller will be compared with the proposed optimal controller. For ease of reference in the following, LS and GAHAM will represent the switching PID control based on linear identified models and the proposed optimal PID control derived through genetic algorithm optimization of a PID controller based on Hammerstein models, respectively. In every situation, including cases involving

square signals, the reference position has undergone filtration via an LPF. Had a perfectly square signal been employed, the input voltage could potentially exceed the maximum motor voltages. An LPF is not necessary for square reference stiffness since the change in stiffness only modifies the voltage difference, not the voltage itself.

### 5.1. Square Position and Stiffness Tracking

For the initial experiment, square signals are utilized as reference position and stiffness waves. This scenario is encountered when a sudden change in position and stiffness is intended. As an example, in an exoskeleton for rehabilitation, it may be required to adjust the stiffness or position very fast to keep the balance of the body [46]. A square signal with an amplitude of 30° and a frequency of 10 s is designated as the reference position. The designated range for reference stiffness spans from 10 to 40 Nm/rad, also with a frequency of 10 s. It is assumed that there exists a phase delay between the reference position and stiffness, facilitating the examination of their individual effects on the system. The experimental outcomes are depicted in Figures 14 and 15. Figure 14 displays position measurements, revealing that the LS method results in oscillatory motion, particularly noticeable at lower stiffness levels. This behavior is likewise evident in voltage measurements for the LS method. While a decrease in stiffness from 40 Nm/rad to 10 Nm/rad introduces a small error in the GAHAM method, this error diminishes over time, approaching zero. The PID gains for the GAHAM method are depicted in Figure 15, demonstrating smooth adjustments corresponding to variations in reference position and stiffness values.

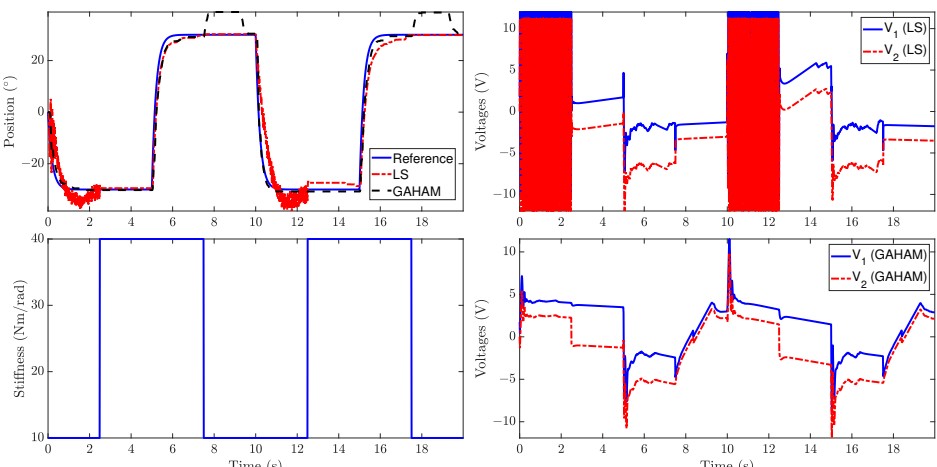

**Figure 14.** Position and stiffness tracking together with motors' voltages for LS and GAHAM methods subject to square position and stiffness waves.

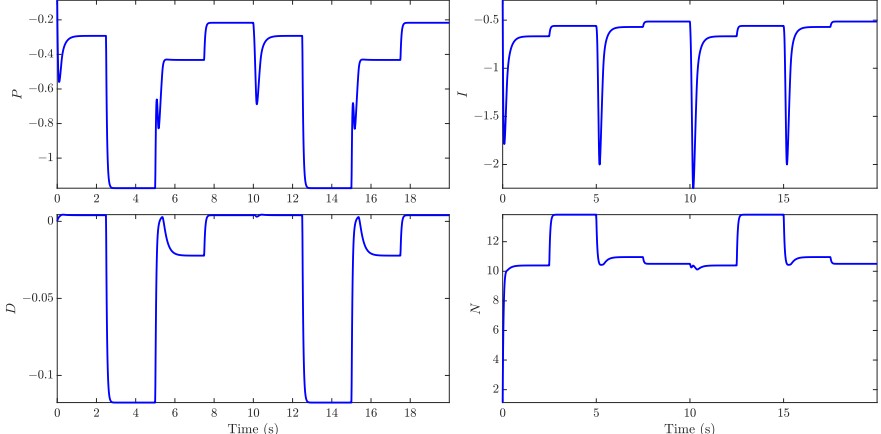

**Figure 15.** Proportional (*P*), integral (*I*), and derivative (*D*) gains and filter parameter (*N*) for GAHAM method subject to square position and stiffness waves.

### 5.2. Sinusoidal Position and Stiffness Tracking

Assume sinusoidal waves applied for both position and stiffness, which is a common movement for rehabilitation purposes. As an example, sinusoidal movements have been used in [47] to treat movement impairments for the upper limb. The amplitude and frequency of a reference position are supposed to be 45° and 10 s, respectively. The reference stiffness starts from 5 Nm/rad and increases to the maximum value of 50 Nm/rad and decrease again to its minimum of $t = 20$ s. From the experimental results shown in Figures 16 and 17, the LS and GAHAM methods have approximately the same position and stiffness tracking performance. The voltage difference for both methods starts with a minimum value and increases to a maximum value of $t = 10$ s and decreases again to its minimum value of $t = 20$ s. PID gains in Figure 17 also confirm that the interpolated gains are adequately produced according to the stiffness value and reference position amplitude.

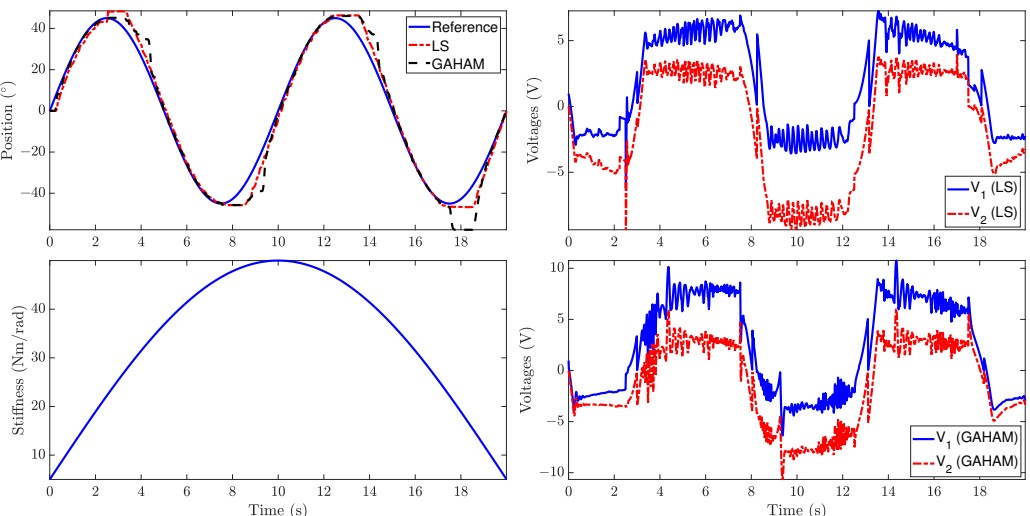

**Figure 16.** Position and stiffness tracking together with motors' voltages for LS and GAHAM methods subject to sinusoidal position and stiffness waves.

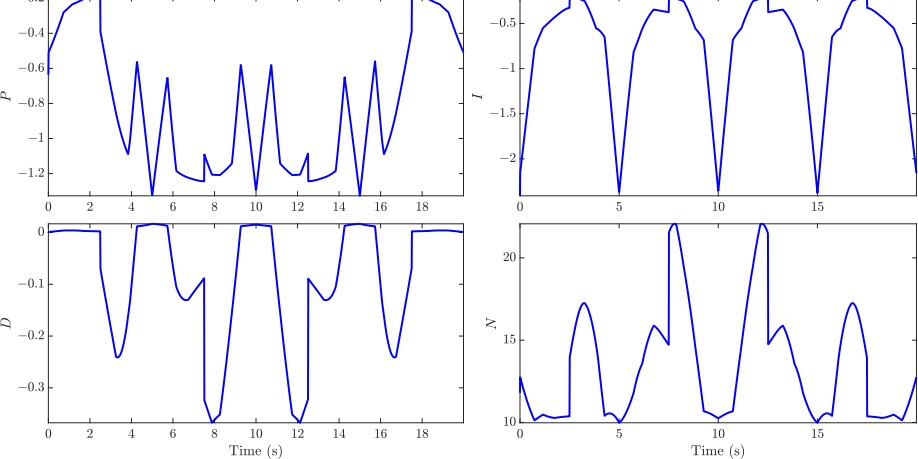

**Figure 17.** Proportional ($P$), integral ($I$), and derivative ($D$) gains and filter parameter ($N$) for GAHAM method subject to sinusoidal position and stiffness waves.

### 5.3. Sawtooth Position and Sinusoidal Stiffness Tracking

In this experiment, a sawtooth signal with an amplitude of 70° and a period of 10 s was used to represent the reference position, while a sinusoidal wave with a period of 40 s and an amplitude ranging from 5 Nm/rad to 50 Nm/rad was employed to represent the reference stiffness. The results are depicted in Figures 18 and 19. This scenario represents a

more realistic scenario in body movement, which is the combination of gradual and abrupt changes of the position.

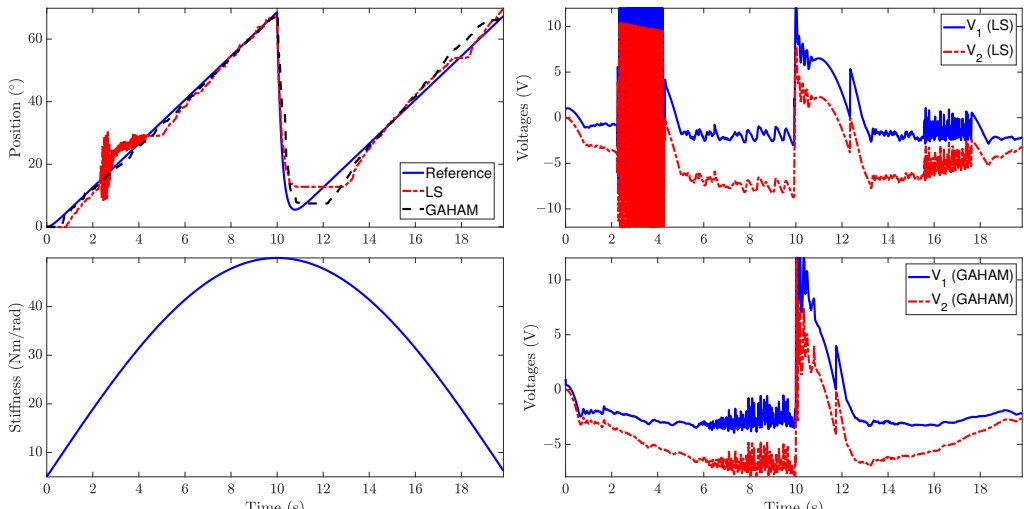

**Figure 18.** Position and stiffness tracking together with motors' voltages for LS and GAHAM methods subject to sawtooth position and sinusoidal stiffness waves.

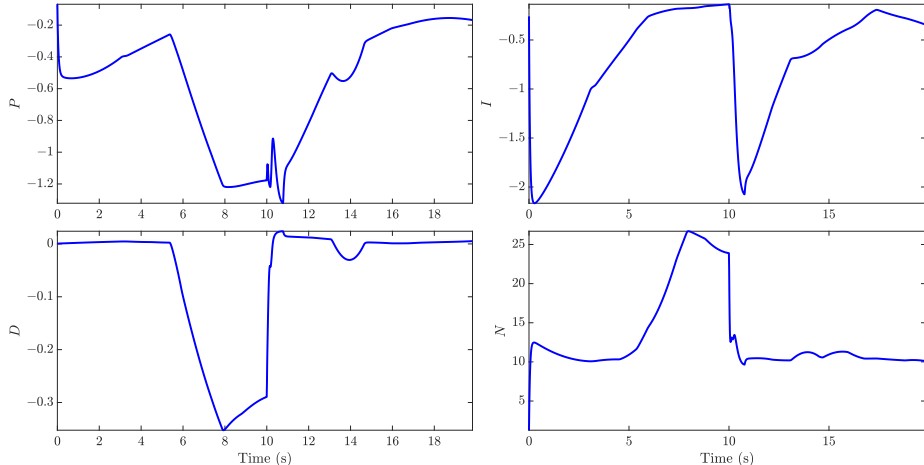

**Figure 19.** Proportional ($P$), integral ($I$), and derivative ($D$) gains and filter parameter ($N$) for GAHAM method subject to sawtooth position and sinusoidal stiffness waves.

From Figure 18, it is evident that the proposed method clearly outperforms the LS method in position tracking. This improvement occurs because, as the reference stiffness increases from 5 to 50 Nm/rad, a switching point is reached among the PID controllers in the LS method, resulting in poor performance in position tracking. This phenomenon is also observable in the motors' voltages for the LS method, which exhibit high-frequency oscillations. In contrast, the voltage difference between $V_1$ and $V_2$ for the GAHAM method is commensurate with the reference stiffness profile, ensuring accurate stiffness tracking. Oscillations were only observed in the increasing trend of reference stiffness. Figure 19 illustrates the proportional, integral, and derivative gains, along with the derivative filter parameter of the GAHAM method for the current scenario. As expected from Figure 12, the amplitude of the proportional gain increases for higher stiffness values, particularly around $t = 10$ s. Additionally, during the time interval $[0, 10$ s$]$, as the reference position increases, the amplitude of the integral gain decreases. However, after $t = 10$ s, upon resetting the reference position, the absolute value of the integral gain increases once again. Furthermore, as the reference stiffness approaches its maximum, the amplitude of the derivative gain increases to its peak value.

### 5.4. Square Position and Sawtooth Stiffness Tracking

To verify the performance of the GAHAM and LS methods under a sawtooth wave stiffness reference, a square wave with an amplitude of 45° and a period of 10 s is chosen as the position reference, and a sawtooth signal with a period of 10 s and an amplitude ranging from 5 Nm/rad to 50 Nm/rad is selected as the reference stiffness profile. The obtained results are illustrated in Figures 20 and 21. As expected, perfect tracking is achieved for the GAHAM method even with varying stiffness over the entire range from 5 to 50 Nm/rad. Motors' voltages confirm that the voltage difference (and hence link stiffness) follows the reference sawtooth stiffness. As stiffness increases, the switching from one PID to another leads to poor tracking of the reference position for the LS method. Figure 21 illustrates the $P$, $I$, $D$, and $N$ values for the current experiment. It is obvious that these gains are not changed sharply due to the existence of LPF in the loop, as shown in Figure 13.

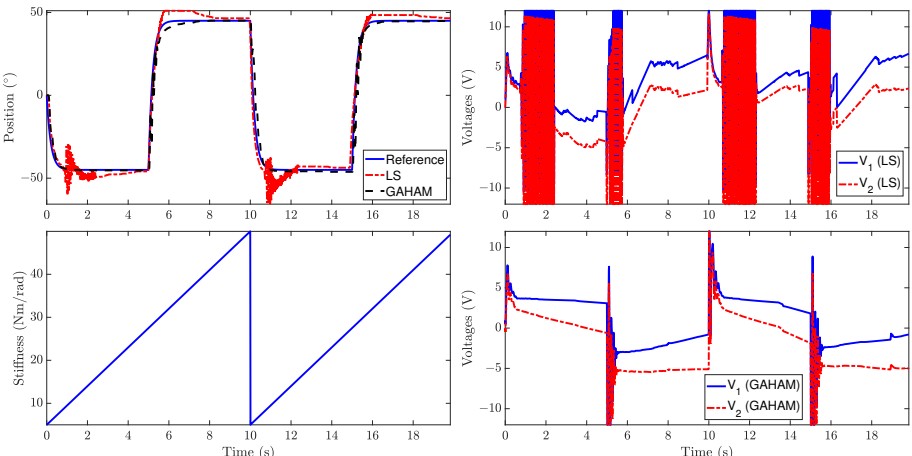

**Figure 20.** Position and stiffness tracking together with motors' voltages for LS and GAHAM methods subject to square position and sawtooth stiffness waves.

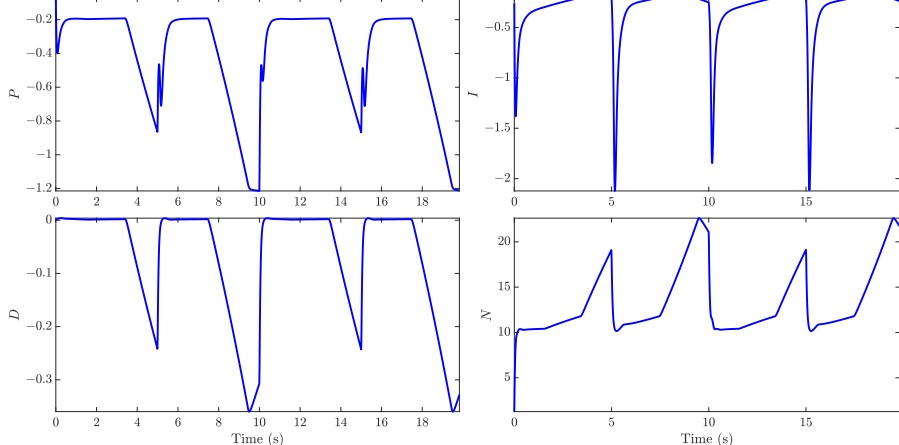

**Figure 21.** Proportional ($P$), integral($I$), and derivative ($D$) gains and filter parameter ($N$) for GAHAM method subject to square position and sawtooth stiffness waves.

### 5.5. Disturbance Rejection Property

To further analyze the performance of the proposed design, we investigate its disturbance attenuation capabilities by subjecting it to a square disturbance with an amplitude of 12 N and a duration of 400 ms. The experimental results are depicted in Figure 22. As shown in the lower plot, the reference stiffness starts at 5 Nm/rad and increases to 20 Nm/rad at $t = 20$ s, before further increasing to 40 Nm/rad at $t = 40$ s. Both the LS and GAHAM methods demonstrate the ability to attenuate the negative effects of the disturbance within

a limited time frame, with minor differences in performance. As expected, a lower stiffness setting results in larger oscillations compared with a higher stiffness value.

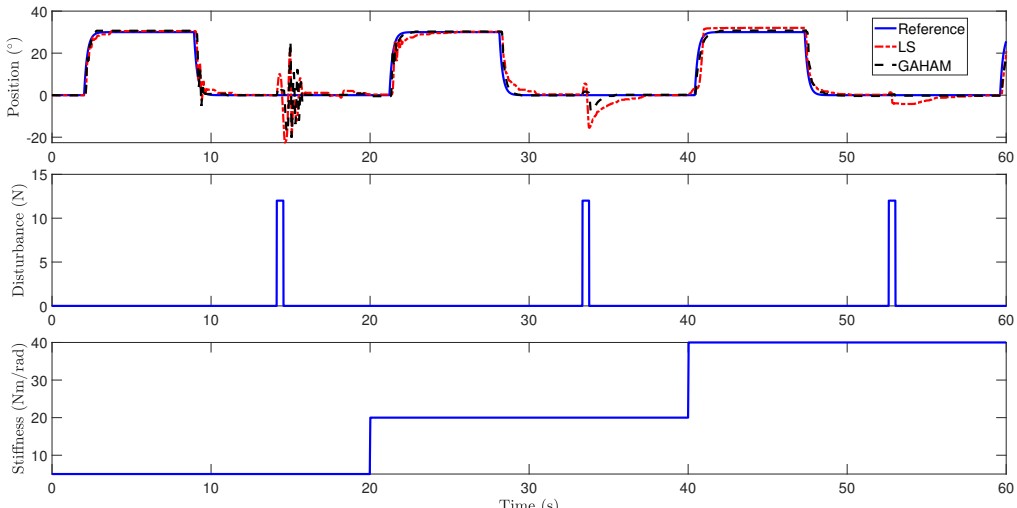

**Figure 22.** Position and stiffness tracking together with applied disturbance for LS and GAHAM methods subject to square position and incremental stiffness waves.

While the current scenario is presented to evaluate the robustness property of the proposed design, its robustness to external disturbances or model uncertainties is not explicitly investigated. To the best of the authors' knowledge, no robust controller has been introduced for multiple Hammerstein models in the literature. This subject poses a challenging task and remains the focus of ongoing research. However, some existing robust controllers introduced for single Hammerstein models can serve as a starting point [48].

## 6. Discussion

The proposed control system design offers several advantages over the existing model-based design discussed in the literature. As previously mentioned, there is no necessity for measuring or estimating system parameters, a task that can sometimes be challenging or impractical. Modeling nonlinear effects like friction, backlash, or dead zones is difficult, and certain other effects remain unknown for modeling purposes. With the identification approach, the system is treated as a black box, requiring no further detailed information. The only essential information needed is the amplitude of the stimulating PRBS signal and the minimum variation time [37]. The minimum variation time refers to the shortest time interval within which the system responds to the input signal. This information can be easily obtained through straightforward experiments for any system.

Different from the methods in the literature, stiffness was produced using the voltage difference method here. As its advantage, this method is easy to implement and requires simple experiments. Additionally, no stiffness model is necessary, and only a force sensor or load cell is required for torque measurements. These properties enable us to apply this method to other types of mechanisms, such as pneumatic and electrohydraulic actuators. The proposed pure data-driven approach enhances the modeling procedure and control system design over the existing model-based methods extensively used in the literature. Furthermore, the voltage difference method used for stiffness production precludes the modeling process for link stiffness.

Many of the current approaches necessitate the measurement of both motors and link positions. Even in studies employing identification methods such as [29], motor positions must be measured. The suggested design offers an advantage by relying solely on link position measurements, requiring only a single position sensor (encoder) for the link. Consequently, the proposed design simplifies the setup and reduces the overall sensor requirements.

One of the challenges of the data-driven methods is the limited range of the link position (in the current setup [−90°, 90°]). When the PRBS signals are being applied to the motors, there is no control on the range of possible movements of the link. Although some limit switches have been included in the hardware to avoid physical damage to the hardware, if the link position for a specific PRBS signal violates the allowed range, then the measured data will not be valid for the identification process. To avoid such situations, the PRBS duration was restricted in time. As an example, for the case of $\Delta V = 3$ V, the final time of a PRBS signal was limited to 10 s, as shown in Figure 6.

Enhancing the identification process could involve the utilization of APRBS rather than PRBS signals [37]. Unlike fixed-amplitude PRBS signals, APRBS offers varying amplitude levels. The variability in amplitude within APRBS signals is advantageous for effectively stimulating nonlinear systems, such as the AVSA, compared with PRBS signals with constant amplitudes. Regrettably, employing APRBS signals in the AVSA setup is unfeasible due to the risk of exceeding the physical limitations of the load, particularly within the range of [−90°, 90°]. This constraint arises from the hardware configuration.

The findings presented in Section 3 confirmed the superior performance of Hammerstein models in capturing nonlinear phenomena within the AVSA system, such as gearbox friction in the motors. Improved models are crucial for enhancing system control. For the optimization process, resolutions of voltage difference (stiffness) and position levels were set at 1 V and 10°, respectively. However, these values can be reduced for enhanced performance. Decreasing the resolutions leads to a rapid increase in the number of optimizations required by the genetic algorithm. Specifically, the number of optimizations needed is equal to the product of the number of stiffness levels and the number of reference position levels. While higher numbers of position and stiffness levels amplify the computational load, they facilitate finer interpolation among the optimized gains, resulting in improved control. These resolutions were determined through a trial-and-error process to achieve satisfactory responses from the AVSA setup.

The voltage difference method offers a versatile approach to stiffness control in actuators, including those with antagonistic structures. Regardless of the actuation source (electric motors, hydraulics, pneumatics, shape memory alloys, cable-driven systems, or magnetic fields), the voltage difference method can be applied to adjust stiffness. In simpler terms, this method allows for the independent control of output stiffness based on voltage difference, without being influenced by link position, actuator complexity, or desired stiffness range. However, one key consideration is the need for non-backdrivable actuation units. In the presented design, two motors with worm gearboxes ensure that the rotation of one motor does not affect the other, enabling stiffness control through voltage difference. This non-backdrivable property is crucial for applying the voltage difference method to other types of AVSAs. While worm gearboxes achieve this in electromechanical designs, alternative solutions are necessary for different AVSA types to achieve the same behavior.

While initially introduced and applied to an AVSA setup, the proposed design can be adapted for use with other types of VSAs, such as SVSAs. It is commonly understood that, in SVSA configurations, a larger motor is dedicated to position control, while a smaller one is employed for stiffness adjustment. Consequently, various stiffness levels can be generated by adjusting the voltage levels of the stiffness (smaller) motor. This approach allows for the establishment of a relationship between the voltage applied to the stiffness motor and the resulting stiffness, without necessitating detailed knowledge of the structural dynamics. Furthermore, for each stiffness level, suitable PRBS signals can be applied to the position motor to identify the position dynamics of the SVSA.

The suggested design is suitable for a single-degree-of-freedom AVSA with one link at the output. For AVSAs with two or more links, various configurations are possible. One potential configuration involves connecting two AVSA units in series, with one AVSA unit (unit A) serving as the load on the link of another AVSA unit (unit B). If the intention is to apply the proposed design to this two-link AVSA, then unit A can be controlled using the same method as outlined earlier. Since unit B carries a load on its link (unit A), the identifi-

cation process must be conducted considering the existence of this load, which is unit A. Consequently, identified models for units A and B will inherently differ. Subsequently, PID controllers can be individually designed for each unit based on their identified models to achieve position tracking. In essence, although PID controllers are independently designed for each unit, the identification process varies for the two units. The design of stiffness control requires more careful consideration. The stiffness at the output link of unit A results from the combination of individual stiffness values for each unit. One potential method for measuring the resulting stiffness at the output involves considering several VDs (e.g., five VDs, as in the proposed design) for each unit. This approach yields 25 different levels of output stiffnesses by exploring all possible combinations of VDs for the two units. For each VD combination, a load cell can be employed to measure the stiffness at the output link. Subsequently, the measured stiffness values, corresponding to the VDs of the two units, can be fitted by a surface. While a single unit involves a straightforward curve fitting, as depicted in Figure 5, the case of two units extends to a surface fitting with two VD variables. The same logic can be exploited for higher degrees of freedom.

## 7. Conclusions

A novel optimal PID controller was introduced for the position and stiffness control of AVSA utilizing Hammerstein models. The voltage difference method was employed to convert the original two-input, two-output system into a single-input, single-output system. For each stiffness level, a Hammerstein model was identified using the least squares method. Experimental results demonstrated an improvement of approximately 90% for Hammerstein models compared with linear models. Genetic algorithms were utilized to compute the optimal PID gains for various position and stiffness values. Linear interpolation of the obtained optimal gains ensures precise position tracking and rapid stiffness transition. Comparison between the GAHAM method and the LS method across multiple scenarios confirms the capability of the proposed approach in facilitating smooth transitions of stiffness and achieving superior position tracking performance. The minimum and maximum achievable stiffness values with the designed hardware is 5 Nm/rad and 50 Nm/rad, respectively. Considering a link length of 9.175 cm, the maximum and minimum load weights for making a deflection of $1°$ at the endpoint are 9.5 N and 0.95 N, respectively. These values fall in a range for which the user has an intuition of the stiffness quantity by applying external force on the link endpoint. The rise time of the position tracking was approximately 1 s, and the operating region of the link was $[-90°, 90°]$. With the proposed method, position and stiffness can be controlled simultaneously, and there is no need for theoretical modeling and parameter estimation/measurement. This property lets us design a control system for any AVSA without any prior information about the hardware characteristics.

**Author Contributions:** Conceptualization, A.J., H.H. and R.C.; methodology, A.J., H.H. and R.C.; software, A.J. and H.H.; validation, A.J., H.H., K.P. and R.C.; formal analysis, A.J.; investigation, A.J. and R.C.; resources, H.H. and R.C.; data curation, A.J. and K.P.; writing—original draft preparation, A.J. and R.C.; writing—review and editing, A.J. and R.C.; visualization, A.J.; supervision, R.C.; project administration, R.C.; funding acquisition, R.C. All authors have read and agreed to the published version of the manuscript.

**Funding:** This research project was partly funded by the Ratchadapiseksompotch Fund of Chulalongkorn University and by the National Research Council of Thailand.

**Data Availability Statement:** The logged position measurements together with the resulting linear and Hammerstein models for all voltage differences are freely available in the Mendeley database accessed on 6 February 2024: https://data.mendeley.com/datasets/vbdhn43c8y/1.

**Acknowledgments:** This research project is supported by the Second Century Fund (C2F), Chulalongkorn University.

**Conflicts of Interest:** The authors declare no conflicts of interest.

## Abbreviations

| | |
|---|---|
| APRBS | Amplitude-modulated Pseudo Random Binary Sequence |
| AVSA | antagonistic variable stiffness actuator |
| CW | clockwise |
| CCW | counterclockwise |
| EMAVSA | electromechanical antagonistic variable stiffness actuators |
| FAVSA | fluidic antagonistic variable stiffness actuators |
| GA | genetic algorithm |
| HIL | hardware in the loop |
| LPF | low-pass filter |
| MIMO | multi input multi output |
| MPC | model predictive control |
| PEA | parallel elastic actuator |
| PID | proportional integral derivative |
| PRBS | pseudo random binary sequence |
| PSO | particle swarm optimization |
| RMSE | root mean square error |
| SA | simulated annealing |
| SEA | serial elastic actuator |
| SISO | single input, single output |
| SMA | shape memory alloy |
| SMC | sliding mode control |
| SVSA | serial variable stiffness actuator |
| VD | voltage difference |
| VSA | variable stiffness actuator |

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
