# Peer review of "Data-Driven Position and Stiffness Control of Antagonistic Variable Stiffness Actuator Using Nonlinear Hammerstein Models"

_jsan, doi:10.3390/jsan13020029_

Round 1

Reviewer 1 Report

Comments and Suggestions for Authors

The article is very interesting. It contains a proposal for modeling an actuator with non-linear stiffness characteristics. The issue is very interesting and "up-to-date" because it concerns modeling the movements of robot actuators so that they have characteristics similar to those of living organisms, i.e. they move smoothly with appropriate dosing of forces, speeds and accelerations. The entire article is a study approach that is the basis for possible further application work.

The authors postulate that their proposed use of the Hammerstein-Wiener model allows for better representation of nonlinearities related to spring characteristics, friction effects and dead zones than the linear model. The authors mention the implementation of stiffness through the voltage difference of the motors, and not through their positioning, as the elements of novelty. They also highlight the implementation of the Hammerstein-Wiener model as the basis for modeling and the identification of PID gain coefficients using a genetic algorithm based on predefined stiffnesses and positions.

The test site has been carefully described, both schematically and in detail, in terms of physical implementation. The solution used is interesting because it allows you to enter both the base voltage and the voltage difference between the motors as controlled variables, which allows you to independently control the stiffness and output position. The authors also noticed differences between theoretically identical motors and gears, resulting in different behavior in clockwise and counterclockwise directions (Fig. 6).

In the analytical part devoted to model identification, 5 linear models and the non-linear Hammerstein-Wiener model were considered. Fig. 10 shows the discrepancies between the linear models and the Hammerstein model and the real course. The error values for the linear models were ten times higher than for the Hammerstein-Wiener model. The entire complex system of switching PID controllers depending on the decision is shown in Fig.11. The structure was then optimized using a genetic algorithm in 45 combinations including 9 amplitude variants and 5 stiffness variants (H-W models).

The whole thing was subjected to experimental tests verifying the modeling accuracy by using 4 different signal variants, and the sensitivity to interference was additionally tested.

Remark:

1. I strongly suggest moving the list of abbreviations to the beginning of the article. This will make it much easier for readers to read it.

Reviewer 2 Report

Comments and Suggestions for Authors

This paper gives an investigation into the development and application of an optimal PID controller for an antagonistic variable stiffness actuator (AVSA) based on nonlinear Hammerstein models. By employing the voltage difference method, the authors have systematically addressed the challenges associated with modeling and controlling AVSAs, which offers novel insights into the dynamics and control strategies of such systems.

Generally speaking, a revision has to be prepared. Some questions and comments are raised below:

- The paper thoroughly elaborates on the technical aspects and innovations introduced in the AVSA control strategy. However, it would be beneficial to discuss the potential limitations or challenges that might arise when scaling this methodology for different types of AVSAs or in varied application contexts. How does the proposed control system adapt to actuators with significantly different dynamics or in applications with varying stiffness requirements? 

- In the experimental setup and control design sections, the authors have effectively utilized a genetic algorithm for optimizing PID gains, considering various stiffness levels and reference position amplitudes. Could the authors elaborate on the choice of the genetic algorithm over other optimization techniques? Specifically, how does the GA compare in terms of computational efficiency and effectiveness in finding optimal gains with other methods like particle swarm optimization (PSO) or simulated annealing (SA)?

- The experimental results section provides a compelling argument for the proposed method's efficacy. However, the robustness of the control system against disturbances or model uncertainties is not explicitly addressed. Could the authors comment on the system's robustness and any strategies implemented or recommended to enhance its resilience against such factors?

- The document exhibits a high level of technical detail and clarity, making it accessible to readers familiar with the subject matter. Nonetheless, there are minor typographical errors and instances where the consistency of terminology could be improved. Ensuring consistency in terms like "AVSA," "PID," and "GA" throughout the text would enhance readability.

- In the section of experimental results, the visual representation of data in Figures 15–20 effectively illustrates the performance comparison between the proposed GAHAM method and the LS method. It is recommended to include a brief description of each figure within the text to guide readers through the results more intuitively.

Comments on the Quality of English Language

Minor improvement.

Reviewer 3 Report

Comments and Suggestions for Authors

The research deals with applying nonlinear Hammerstein models to ensure data-driven positioning of actuators. The title is topical and of practical interest. However, the following flaws should be eliminated before further proceedings of the manuscript:

1. It is unclear where the Research Methodology starts.

2. An approach (17)–(19) is well-known as a Moore–Penrose pseudoinverse. However, it was not mentioned in the text.

3. Due to many already published articles on AVSAs, the authors should highlight the scientific novelty of their methodology and obtained results.

4. Each subsection (e.g., Control Design, Experimental Research) is based on the other studies. Therefore, it is unclear which significant developments the researchers made.

5. The number of self-citations is inappropriate: [1–5, 11, 14, 20, 44]. I highly recommend decreasing this level to 1–2 items.

Overall, the manuscript needs improvements.

Comments on the Quality of English Language

Minor editing of the English language is recommended.

Reviewer 4 Report

Comments and Suggestions for Authors This is a comprehensive work on PID controllers with detailed analysis. GAHAM method and the LS method are compared for multiple cases with consistent results. Minor revision is suggested with following comments:  

1. It would be good to combine Figure 1 and 2 with labels matching each component.

2. How is Eq. (1) obtained and what is the accuracy of the fitting curve? Same for Eq. (4) (5).   3. Are there any repeated measurements for real data?

Round 2

Reviewer 3 Report

Comments and Suggestions for Authors

since the article has been improved, you can make a decision to publish it without changes.

Comments on the Quality of English Language

Average.